# Impact of Positive Radial Margin on Recurrence and Survival in Perihilar Cholangiocarcinoma

**DOI:** 10.3390/cancers14071680

**Published:** 2022-03-25

**Authors:** Francesco Enrico D’Amico, Claudia Mescoli, Silvia Caregari, Alessio Pasquale, Ilaria Billato, Remo Alessandris, Jacopo Lanari, Domenico Bassi, Riccardo Boetto, Francesco D’Amico, Alessandro Vitale, Sara Lonardi, Enrico Gringeri, Umberto Cillo

**Affiliations:** 1Department of Surgical, Oncological and Gastroenterological Sciences (DiSCOG), University of Padova, 35128 Padua, Italy; silvia.caregari@gmail.com (S.C.); alessiopasquale@gmail.com (A.P.); remo.alessandris@aopd.veneto.it (R.A.); jacopo.lanari@unipd.it (J.L.); alessandro.vitale@unipd.it (A.V.); enrico.gringeri@unipd.it (E.G.); cillo@unipd.it (U.C.); 2Second General Surgical Unit, Padova Teaching Hospital, 35128 Padua, Italy; ilaria.billato@aopd.veneto.it (I.B.); domenico.bassi@aopd.veneto.it (D.B.); riccardo.boetto@aopd.veneto.it (R.B.); francesco.damico@aopd.veneto.it (F.D.); 3Department of Medicine, Surgical Pathology and Cytopathology Unit, University of Padova, 35128 Padua, Italy; claudia.mescoli@aopd.veneto.it; 4Medical Oncology 1 Unit, Department of Oncology, Veneto Institute of Oncology IOV-IRCCS, 35128 Padua, Italy; sara.lonardi@iov.veneto.it

**Keywords:** cholangiocarcinoma, perihilar cholangiocarcinoma, resection status, margin, extrahepatic bile duct, major hepatectomies

## Abstract

**Simple Summary:**

The only potentially curative treatment of perihilar cholangiocarcinoma (PHC) is complete (R0) resection. This is difficult to achieve and great effort should be made to optimise surgical margins assessment and to thoroughly define their prognostic value. When considering resections for PHC, not only bile duct margins (ductal margins, DM), but also the liver transection plane and the dissection plane in the hepatoduodenal ligament (radial margins, RM) should be examined. Studies concerning PHC resections with comprehensive analyses of the recurrence and survival related to margins status most frequently consider only ductal margins. The importance of also assessing radial margins’ prognostic value was recently introduced and deserves to be further studied. To our knowledge, there is currently no evidence of prognostic value of isolated positive RM. Therefore, the aim of this study was to evaluate the incidence and to investigate the effects on the recurrence and survival of positive isolated RM in resected PHC.

**Abstract:**

In resected perihilar cholangiocarcinoma (PHC), positive ductal margin (DM) is associated with poor survival. There is currently little knowledge about the impact of positive radial margin (RM) when DM is negative. The aim of this study was to evaluate the incidence and the role of positive RM. Patients who underwent surgery between 2005 and 2017 where retrospectively reviewed and stratified according to margin positivity: an isolated RM-positive group and DM ± RM group. Of the 75 patients identified; 34 (45.3%) had R1 resection and 17 had positive RM alone. Survival was poorer in patients with R1 resection compared to R0 (*p* = 0.019). After stratification according to margin positivity; R0 patients showed better survival than DM ± RM-positive patients (*p* = 0.004; MST 43.9 vs. 23.6 months), but comparable to RM-positive patients (*p* = 0.361; MST 43.9 vs. 39.5 months). Recurrence was higher in DM ± RM group compared to R0 (*p* = 0.0017; median disease-free survival (DFS) 15 vs. 30 months); but comparable between RM and R0 group (*p* = 0.39; DFS 20 vs. 30 months). In univariate and multivariate analysis, DM positivity resulted as a negative prognostic factor both for survival and recurrence. In conclusion, positive RM resections appear to have different recurrence patterns and survival rates than positive DM resections.

## 1. Introduction

Cholangiocarcinoma accounts for approximately 10% of all hepatobiliary tumours and perihilar cholangiocarcinoma (PHC) represents two-thirds of cases [1]. Surgical resection is the only potentially curative treatment for patients with PHC.

The main goal of surgery is a complete (R0) resection, which is difficult to achieve. In different studies, its rates are reported in a wide range, between 50 and 90% [2]. Since surgical margin positivity is a relevant prognostic factor for recurrence and survival, careful margins assessment is essential and should be thoroughly analysed.

Relapse rates are high despite curative intent surgery, thus, adjuvant therapies also play an important role in the management of this disease [3]. SWOG S0809 phase II trial established the promising efficacy of adjuvant chemotherapy followed by adjuvant chemoradiation for biliary tract cancer [4]. Later on, since 2017, the results of randomised phase III clinical trials were reported. In particular, the BILCAP trial, despite failing to meet its primary endpoint by intention-to-treat analysis, led to the wide adoption of capecitabine as an adjuvant treatment. Nonetheless, the role of adjuvant systemic chemotherapy is still an object of debate and the results of several phase I to III clinical trials currently evaluating the role of novel therapeutic approaches are highly awaited [5].

When considering resections for PHC, not only bile duct margins (ductal margins, DM) but also the liver transection plane and the dissection plane in the hepatoduodenal ligament (radial margins, RM) should be examined and taken into account. This important concept was highlighted by Shinohara et al. in an innovative study on radial margin status in resected PHC published in 2021 [6].

There is currently little knowledge about the impact of isolated positive RM as most studies focus only on the impact of DM in resected PHC. The aim of this study is to retrospectively evaluate the incidence of positive RM in our high-volume institution and to investigate the impact of isolated positive RM on the recurrence and survival in resected PHC.

## 2. Materials and Methods

### 2.1. Study Population

All consecutive patients resected for PHC with curative intent at Padova University Hospital—Hepatobiliary Surgery and Liver Transplantation Unit between December 2005 and December 2017 were retrospectively reviewed. Demographic, clinical, and treatment-related variables were recorded. These are sex, age, comorbidities, main serological data (platelet count, prothrombin time, INR, total bilirubin, creatinine, sodium, albumin, CA19-9, CEA, AFP), tumour characteristics, and perioperative procedures.

Preoperative imaging was always performed, including chest, abdomen, and pelvic computed tomography (CT) and magnetic resonance cholangiopancreatography (MRCP); endoscopic ultrasound (EUS) and positron emission tomography (PET) CT were performed in selected patients. The extension of the disease was classified according to the Bismuth-Corlette classification [7].

A brushing cytology diagnosis was attempted in all patients who had been preoperatively drained; negative results were not considered to be an absolute contraindication to surgery.

### 2.2. Preparation for Surgery and Surgical Procedure

Preoperative biliary drainages or stents were placed percutaneously during ultrasound-guided procedures or by endoscopic retrograde cholangiopancreatography (ERCP) in patients suffering from obstructive jaundice, cholangitis, or dilation of the bile duct in the liver remnant.

Staging laparoscopy was performed only in selected cases. Intraoperative ultrasound was always performed to exclude liver metastases, evaluate tumour extension and vascular involvement, and guide transection plane assessment. During surgery, proximal and distal ductal margins were routinely submitted for frozen-section analysis, leading to additional bile duct resection, when positive and feasible. In case of tumour encasement, portal vein was resected and reconstructed with or without peritoneal patch or graft interposition. Only one arterial reconstruction was necessary and performed in this series. Regional lymphadenectomy was performed systematically, including hilar, pericholedochal, periportal, peripancreatic, and common hepatic artery lymphnodes. Lymphadenectomy was classified according to the Japanese Gastric Cancer Association (JGCA) [8].

Liver resections were classified according to the Brisbane 2000 terminology [9]. Postoperative complications were evaluated based on the Clavien-Dindo classification of surgical complications [10].

Adjuvant systemic chemotherapy was administered according to NCCN guidelines [11] and patients’ ECOG status.

### 2.3. Pathologic Evaluation

All surgical specimens were submitted for histopathological evaluation by adopting a standardized protocol where only dedicated pathologists with expertise in gastrointestinal pathology assessed the specimens and drafted the reports. To adequately determine residual disease status, five resection planes (common bile duct, proximal bile duct, hepatic artery, portal vein, and liver parenchyma) and one periductal dissection plane were obtained from each surgical specimen. Histological reports included the following clinicopathological parameters: tumour’s size and differentiation grade, vascular, perineural and intraductal invasion, resection/dissection planes status, tumour growth pattern (mass forming, periductal infiltrating, intraductal, and mixed type), lymph node status and coexisting pathology (inflammation, fibrosis, steatosis, sclerosing cholangitis etc.).The annular resection margins (common bile duct, proximal bile duct, hepatic artery, and portal vein) were recorded as positive or negative, since their slices are generally 1 to 3 mm thick and a “negative” annular plane ensures a margin of > 1 mm. For non-annular planes (periductal and liver parenchyma), margins were assessed as positive if cancer was up to 1 mm along the dissection plane in the hepatoduodenal ligament or on the liver transection plane. The presence of severe dysplasia or carcinoma in situ at the surgical margin was classified as a negative margin.

Data were retrospectively retrieved from the original pathology reports by a single pathologist (C.M.) expert in hepatobiliary malignancy, and the pathological staging was updated according to the revised American Joint Committee on Cancer (AJCC)/Union for International Cancer Control (UICC) TNM staging system 8th edition [12].

### 2.4. Study Design and Statistical Analysis

To evaluate the outcome of patients with positive radial margin and negative ductal margin after curative intent resection, this study population was divided into three groups according to the state of the resection margin:R0 group: patients with negative DM and RMRM group: patients with positive RM and negative DMDM ± RM group: patients with positive DM regardless of RM status

Values for categorical variables were expressed as totals and percentages whereas for continuous variables they were expressed as medians and ranges or interquartile ranges. Statistical analyses were performed using the Pearson’s chi-squared test or Fisher’s test for categorical variables and the Kruskal–Wallis rank sum test for continuous variables.

The length of follow-up was calculated from the date of surgery to the date of patient death (overall survival—OS) or the latest follow-up. The durations of follow-up and survival were expressed as medians (interquartile ranges).

Survival and recurrence curves were calculated using the Kaplan–Meier technique and compared with the log-rank test. Prognostic factors of recurrence and survival were identified through univariate and multivariate analyses using the Cox proportional hazards model.

A *p* value < 0.05 was considered to indicate statistical significance, variables with a *p* value < 0.1 were considered of marginal statistical significance.

Statistical analyses were performed using R, RStudio 4.1.0 (2021).

## 3. Results

### 3.1. Characteristics of the Patients and Surgical Procedures

Between December 2005 and December 2017, 82 consecutive patients diagnosed with PHC were resected with curative intent at Padova University Hospital—Hepatobiliary Surgery and Liver Transplantation Unit. Of these, seven patients were excluded from the study: two were lost at follow-up and five died postoperatively. The remaining 75 patients represent the study population, including a majority of men (69%) with a median age of 65.7 years (60.1–73.5). The patients’ characteristics and surgical procedures are depicted in Table 1.

There was no statistical difference between the groups considering all variables, except for adjuvant chemotherapy administration. DM ± RM group had a higher median CA19-9, with a trend toward a significant difference. According to the Bismuth classification criteria, the majority of patients in this study were diagnosed with type III and IV tumours.

Left hepatectomy was the most frequently performed procedure (42/75, 56%) with a higher incidence of DM ± RM positivity (11/42, 26.2%) than RM positivity (7/42, 16.7%); on the other hand, in right hepatectomies (22/75, 29%), RM positivity was more frequent (7/22, 31.8%) than DM ± RM positivity (3/22, 13.6%). R0 rate was similar in these two types of resections: 57.1% (24/42) in left hepatectomies and 54.5% (12/22) in right hepatectomies.

Patients in the RM group had the highest rate of vascular reconstructions, but without significant differences between groups.

Adjuvant chemotherapy was administered significantly differently in the three groups (*p* = 0.039), being respectively most (15/17, 88%) and least (8/17, 47%) frequently used to treat RM and DM ± RM patients. In the R0 population, 26 patients (26/41, 63%) received adjuvant chemotherapy. Moreover, adjuvant radiotherapy was administered with statistically different rates in the three groups (*p* = 0.010).

### 3.2. Pathologic Features

Of the 75 study patients, 34 (45.3%) had an R1 resection. Among these, 17 (50%) had positive RM alone, 8 (23.5%) had positive DM alone, and 9 (26.5%) had both positive DM and RM. Overall, 26 patients had a positive RM status, whereas 17 had a positive DM status. Pathologic features are summarized in Table 2.

Concerning TNM, most of the patients were diagnosed with a T2b tumour, with a higher percentage of T1-T2a samples in the R0 group and the absence of T3–T4 tumours in the RM group. Nodal involvement did not show a significant difference in the three groups, being more frequently absent in the entire population, except for the DM ± RM patients.

Most of the patients presented vascular and perineural invasion without significant differences between groups.

### 3.3. Survival Analysis

The median follow-up of the whole population was 33.2 months (interquartile range (IQR) 21.4–49.5 months) with statistically significant differences between the 3 groups (*p* = 0.03), the R0 population being the one with the longest median follow-up of 36.7 months (IQR 22.0–57.9 months). In the other two subgroups, the median follow-up was 37.3 months (IQR 24.4–44.6 months) and 22.2 months (IQR 10.8–28.8 months), respectively, in the RM and DM ± RM groups.

At 5 years, 2 patients (2/17, 11.8%) of the DM ± RM group, 6 patients (6/17, 35.3%) of the RM group and 20 patients (20/41, 48.8%) of the R0 group (*p* = 0.029) were alive.

Survival analysis showed statistically significant longer survival (*p* = 0.019) in the R0 group compared to the R1 group (Table 3 and Figure 1).

Upon further analysis, it was found that overall survival was not significantly different between the R0 and RM groups (*p* = 0.361), nor between the RM and DM ± RM groups (*p* = 0.113). A significant difference in survival was found between the R0 and DM ± RM groups (*p* = 0.004) (Table 3 and Figure 2).

Variables that had a significant impact on survival in univariate analysis were positive DM ± RM (HR 2.83), CA19-9 > 40 kU/L (HR 4.33), positive lymphnodes (HR 1.99), stage III-IV (HR 2.11), vascular invasion (HR 2.17), complications (HR 3.94), and recurrence (HR 14.78); G3 (HR 2.87) was of marginal statistical significance (Table 4).

In multivariate analysis, DM ± RM positivity (HR 2.85), CA19-9 > 40 kU/L (HR 4.30), Bismuth IV (HR 5.92), vascular invasion (HR 2.51), and recurrence (HR 6.91) were identified as independent prognostic factors for survival (Table 5).

### 3.4. Recurrence Analysis

Overall, 52 patients (71%) developed recurrence, which was more represented in the R1 (26/34 patients, 76%) than in R0 (26/41 patients, 63%) group. The median disease-free survival (DFS) of the whole population was 21.4 months (range 1.15–91.9), and 30 (range 17.3–69.7) and 16.3 (range 11.4–43) months in R0 and R1 groups (*p* = 0.035), respectively.

The estimated cumulative probability of recurrence was significantly lower in R0 than in R1 patients (*p* = 0.023) (Table 6). Further analysis showed that it was also significantly lower in R0 than in DM ± RM patients (*p* = 0.0017). It was not significantly different in R0 vs. RM comparison (*p* = 0.39), nor in RM vs. DM ± RM comparison (*p* = 0.19) (Table 6 and Figure 3).

Variables that had a significant impact on recurrence in univariate analysis were positive DM ± RM (HR 2.72), CA19-9 > 40 kU/L (HR 2.44), vascular invasion (HR 1.97), and complications (HR 3) (Table 7).

In multivariate analysis, DM ± RM positivity (HR 2.95) and complications (HR 4.20) were identified as independent prognostic factors for recurrence (Table 8).

## 4. Discussion

Surgical resection is the only potentially curative treatment for patients with resectable PHC. It is widely reported in literature that patients with positive surgical margins after resection have an overall and disease-specific survival that is significantly lower than patients with negative margins [13,14]. This was recently confirmed by Liang et al. [15] in a systematic review and meta-analysis on prognostic factors of resectable PHC. The study shows that positive resection margin status (R1/R2 vs. R0) has clinically relevant prognostic value on overall survival (HR 2.34) and on disease-free survival (HR 1.96). For this reason, the assessment of the prognostic significance of surgical margins is relevant.

A great variation exists among different centres concerning the rates of positive margins. Indeed, R0 resection rates can range between 50 and 90% [2]. In our series the rate of positive margins was 45.3, with 54.7% R0 resections. These results are fairly in line with the values found by Mueller et al. in their attempt to identify benchmark cut-offs as reference values for PHC surgery (R0 ≥ 56.7% and R1 ≤ 43%) [16].

In most series about PHC recurrence [17] and PHC margin status after resection [18], the authors focused only on ductal margins. Moreover, in some studies, even when radial margins were taken into account, their isolated positivity, with negative DM, was categorized as R0 resection [19].

The lack of information about the status of radial margins could be explained by the challenging interpretation of the specimens [20]. A French multi-institutional study on the accuracy of pathology reports in PHC patients [21] revealed a frequent shortage of important data and in particular, relevant differences in margin-specific information. Indeed, bile duct margins were assessed in 90% of the specimens, liver margins only in 20%, vascular margins in 13%, and periductal soft tissue circumferential margins in 10%. To improve pathology reports’ accuracy, precise orientation of the specimen and marks for relevant resection planes should be provided by the surgeons, and better communication with the pathologists should be achieved.

The importance of the meticulous handling of the resected specimen by the surgeons was also underlined by Shinohara et al. [6] in 2021. In this innovative study was introduced the importance of radial margins assessment in PHC resections. To our knowledge, RM had only been considered in one previous study published in 2005 by Sakamoto et al. [22]. This study on middle and distal bile duct cancer considered 51 patients who mainly underwent pancreatoduodenectomies; only five patients underwent hemiepatectomies, two of them combined with pancreatoduodenectomy. In Shinohara’s analysis, RM was defined as the presence of tumour cells along the liver transection plane or on the dissection plane in the hepatoduodenal ligament. Positive RM was found to be the most common cause of R1 resection for PHC and had a similarly negative effect on postoperative survival as positive DM. In this study, the RM group consisted of RM-positive patients both with positive or negative DM, whereas the DM group was made up of DM-positive RM-negative patients.

In our study, as already found in Shinohara’s series, R1 resections were attributable in most cases to RM positivity (26/34 RM+ vs. 17/34 DM+); our analysis, though, had a different subgroups division. In many analyses about PHC, the population with R1 resection consists only of DM-positive patients, regardless of the status of radial margins since these are usually not considered. What enriches the analysis, in our opinion, is the evaluation of the outcome of patients with isolated RM positivity. This was the aim of our study and thus guided the choice of group partition.

A similar approach to subgroups partition in patients who underwent R1 resection was adopted by Stremitzer et al. [23] in their analysis on isolated positive circumferential margin in PHC resection. In this study, though, resection margins were differently categorised. Surgical margins were defined as the transection sites (proximal and distal) at the bile duct and the liver parenchyma respectively, whereas the circumferential margin was defined as the interface of the extrahepatic bile duct and the surrounding lymphatic/fatty tissue of the hepatoduodenal ligament. The group of patients with positive surgical margins, regardless of circumferential margin status, was compared to the group of patients with an isolated positive circumferential but negative surgical margin. The main difference with our study is the classification of the liver resection margin, considered as the surgical margin in Stremitzer’s study and as radial margin in our analysis. Indeed, the comparison of these two similarly designed analyses shows comparable survival curves in the three groups.

In our study, patients with R0 resections, compared to patients with positive RM, had statistically similar overall survival and disease recurrence even though the curves were slightly different. On the other hand, DM ± RM-positive patients showed statistically poorer survival and higher recurrence rates compared to the R0 group. Moreover, the positivity of DM ± RM had a negative prognostic value both on survival and recurrence in the univariate and multivariate analyses.

Usually, in clinical practice, DM are routinely submitted during surgery for frozen-section analysis. When there is a positive result, an additional resection is attempted when feasible. Zhang et al. [24] demonstrated in their retrospective study that when the final ductal margin was negative, there was an improvement in long-term outcomes. Conversely, RM are not regularly submitted intraoperatively. Whether RM frozen-section analysis during surgery could improve PHC resection outcomes is still unclear.

Concerning adjuvant treatments, radiotherapy was used only in selected cases and in patients with recurrent disease, through a shared decision-making approach. Its significantly different (*p* = 0.010) administration rates between the three groups could most likely be related to this patient-centred indication. A significant difference between groups (*p* = 0.039) was also found in adjuvant chemotherapy administration, which conversely was indicated on the basis of a protocol-guidelines based approach. This treatment was much more represented in the RM group (15/17, 88%) than in the DM ± RM group (8/17, 47%). We suggest that DM positivity could speculatively be the cause of hepaticojejunostomy complications due to recurrences, thus lowering treatment chances for these patients. In these cases the placement of biliary stents or drainages is needed to palliate jaundice, with the risk of the development of additional procedure-related complications, in particular infections. Indeed, septic events are quite frequent in patients with a biliary drainage in place [25]. Adjuvant chemotherapy administration, even if likely beneficial (HR 0.71—HR 0.97 in our series), did not result as a significant prognostic factor on survival and recurrence in the univariate analysis. This finding, though, should not undercut the realistic benefit of adjuvant systemic therapy, as this treatment was found to be a positive prognostic factor for PHC after curative resection in other studies [15]. The result of our analysis could be biased given the small sample size of the series.

The main limitation of our study is its retrospective nature, which could hide unexpected biases. Moreover, we consider a small sample size of patients, all treated at a single centre. A bias derived from this aspect could be, for example, concerning adjuvant treatments, the patient-based indication for radiotherapy, and the absence of a standardised protocol. On the other hand, a major issue of the muti-institutional analyses in the PHC field is the variability of pathologic evaluation of margins in different institutions. Thus, in our opinion, a strength of our work is the single pathologist review of all the specimens.

## 5. Conclusions

In conclusion, in this single institution study, isolated positive RM resections had statistically similar overall survival and disease recurrence compared to patients with R0 resections. DM-positive patients showed statistically poorer survival and higher recurrence rates compared to the R0 group. It is difficult to make definitive conclusions from our analysis, and further multi-institutional studies with centralised pathology reviews are recommended to confirm our preliminary results.

## Figures and Tables

**Figure 1 cancers-14-01680-f001:**
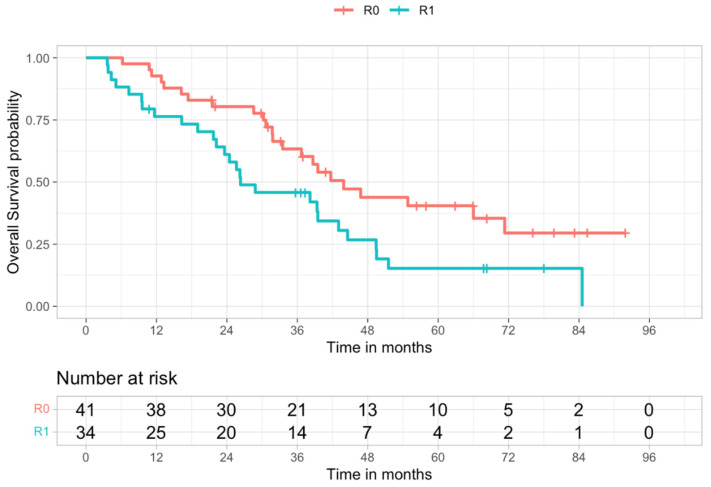
Kaplan–Meier overall survival curves of R0 and R1 populations (log-rank test, *p* = 0.019).

**Figure 2 cancers-14-01680-f002:**
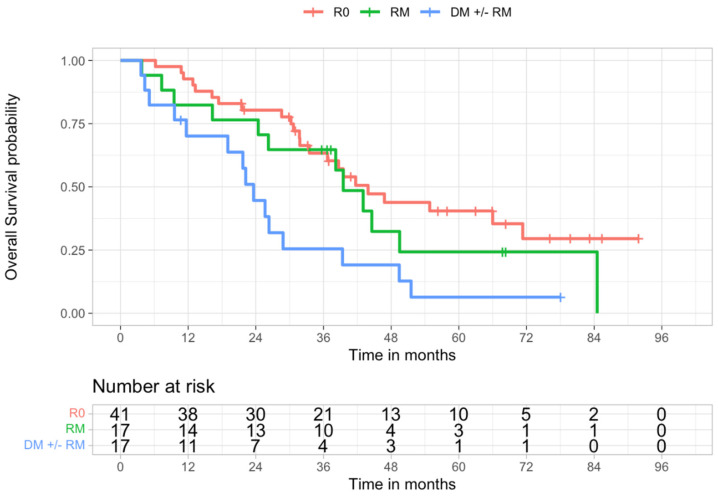
Kaplan-Meier overall survival curves of R0, RM, and DM ± RM populations (log-rank test, *p* = 0.0056).

**Figure 3 cancers-14-01680-f003:**
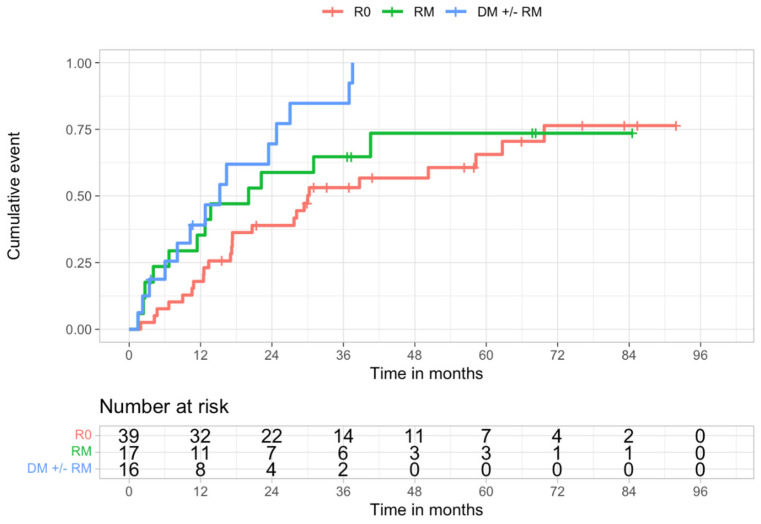
Kaplan-Meier recurrence curves of R0, RM, and DM ± RM populations (log-rank test, *p* = 0.0066).

**Table 1 cancers-14-01680-t001:** Patients’ characteristics and surgical procedures.

	R0 (*n* = 41)	RM (*n* = 17)	DM ± RM (*n* = 17)	Combined (*n* = 75)	*p* Value
Age, median (IQR)	68.6 (62.7–75.6)	63.5 (51.5–68.1)	64.1 (60.3–66.7)	65.7 (60.1–73.5)	0.07
Gender, male	30 (73%)	12 (71%)	10 (59%)	52 (69%)	0.55
Bilirubin (mg/dL), median (IQR)	2.9 (1.0–6.2)	2.1 (0.7–4.2)	1.7 (1.1–5.5)	2.5 (1–5.5)	0.41
CA19-9 (kU/L), median (IQR)	108.0 (38.8–444.6)	312.3 (42.1–1650.0)	612.2 (250.8–2172.0)	243.5 (44.7–1171.5)	0.08
CEA (ng/mL), median (IQR)	2.9 (1.9–3.6)	1.6 (1.2–2.5)	2.2 (1.6–2.9)	2.20 (1.5–3.5)	0.24
Pre-operative drain (PTBD or ERCP)	23 (56%)	12 (71%)	13 (76%)	48 (64%)	0.28
Bismuth type					0.79
I	1 (2.4%)	1 (6.2%)	0 (0%)	2 (2.7%)	
II	7 (17%)	1 (6.2%)	4 (24%)	12 (16%)	
IIIa	13 (32%)	5 (31%)	3 (18%)	21 (28%)	
IIIb	14 (34%)	5 (31%)	7 (41%)	26 (35%)	
IV	6 (15%)	4 (25%)	3 (18%)	13 (18%)	
Surgical procedure					0.68
Left hepatectomy ± S1	24 (59%)	7 (41%)	11 (65%)	42 (56%)	
Right hepatectomy ± S1	12 (29%)	7 (41%)	3 (18%)	22 (29%)	
Central hepatectomy ± S1	1 (2.4%)	1 (5.9%)	0 (0%)	2 (2.7%)	
Isolated bile duct resection	3 (7.3%)	1 (5.9%)	2 (12%)	6 (8%)	
Others	1 (2.4%)	1 (5.9%)	1 (5.9 %)	3 (4%)	
Number of resected segments, median (IQR)	4 (3–4)	4 (4–4)	4 (4–4)	4 (3–4)	0.66
Vascular reconstruction	8 (20%)	6 (35%)	1 (5.9%)	15 (20%)	0.11
Adjuvant chemotherapy	26 (63%)	15 (88%)	8 (47%)	49 (65%)	0.039
Adjuvant radiotherapy	7 (17%)	8 (47%)	1 (5.9%)	16 (21.3%)	0.010

Abbreviations: IQR, interquartile range; CA19-9, carbohydrate antigen 19-9; CEA, carcinoembryonic antigen; PTBD, percutaneous transhepatic biliary drainage; and ERCP, endoscopic retrograde cholangiopancreatography.

**Table 2 cancers-14-01680-t002:** Pathologic features.

	R0 (*n* = 41)	RM (*n* = 17)	DM ± RM (*n* = 17)	Combined (*n* = 75)	*p* Value
Dimension (mm), Median (IQR)	30 (20–42.5)	35 (30–40)	40 (30–40)	30 (22.5–40)	0.37
Grade, median (IQR)	2 (2–2)	2 (2–3)	2 (2–2)	2 (2–2)	0.37
Grade					0.73
1	6 (15%)	1 (5.9%)	1 (7.1%)	8 (11%)	
2	25 (64%)	10 (59%)	10 (71%)	45 (64%)	
3	8 (21%)	6 (35%)	3 (21%)	17 (24%)	
T					0.14
1	6 (15%)	0 (0%)	1 (5.9%)	7 (9.3%)	
2a	14 (34%)	4 (24%)	2 (12%)	20 (27%)	
2b	16 (39%)	13 (76%)	11 (64.7%)	40 (53.3%)	
3	2 (4.9%)	0 (0%)	1 (5.9%)	3 (4%)	
4	2 (4.9%)	0 (0%)	2 (12%)	4 (5.3%)	
x	1 (2.4%)	0 (0%)	0 (0%)	1 (1.3%)	
N					0.52
0	23 (56%)	10 (59%)	8 (47%)	41 (55%)	
+	15 (37%)	4 (24%)	8 (47%)	27 (36%)	
x	3 (7.3%)	3 (18%)	1 (5.9%)	7 (9.3%)	
Stage					0.16
I	7 (17%)	0 (0%)	0 (0%)	7 (9.3%)	
II	16 (39%)	13 (76%)	7 (41%)	36 (48%)	
IIIa	1 (2%)	0 (0%)	0 (0%)	1 (1%)	
IIIb	3 (7%)	0 (0%)	2 (12%)	5 (7%)	
IIIc	10 (24%)	4 (24%)	6 (35%)	20 (27%)	
IV	4 (9.8%)	0 (0%)	2 (12%)	6 (8%)	
Vascular invasion	27 (71%)	14 (82%)	11 (69%)	52 (73%)	0.64
Perineural invasion	30 (81%)	15 (88%)	15 (94%)	60 (86%)	0.60

Abbreviations: IQR, interquartile range; T, tumour (8th AJCC TNM staging); and N, nodes (8th AJCC TNM staging).

**Table 3 cancers-14-01680-t003:** Kaplan-Meier overall survival analysis.

	R0	R1	RM	DM ± RM
1 year	92.7%	76.4%	82.4%	70.1%
3 years	63.3%	45.8%	64.7%	25.5%
5 years	40.5%	15.3%	24.3%	6.4%
Median survival, months	43.9	26.3	39.5	23.6

**Table 4 cancers-14-01680-t004:** Univariate analysis for survival prognostic factors.

	HR	L 95% CI	U 95% CI	*p* Value
RM	1.39	0.69	2.79	0.358
DM ± RM	2.83	1.46	5.48	0.002
CA19-9> 40 kU/L	4.33	1.69	11.07	0.002
N+	1.99	1.09	3.66	0.026
Bismuth IV	1.52	0.71	3.26	0.284
Dimension	1.00	0.99	1.02	0.545
CT adj	0.71	0.39	1.28	0.254
Stage III–IV	2.11	1.19	3.74	0.010
G2	2.39	0.84	6.79	0.104
G3	2.87	0.89	9.21	0.076
Vascular Invasion	2.17	1.04	4.52	0.038
Complications	3.94	1.40	11.04	0.009
Recurrence	14.78	3.57	61.23	<0.001

Abbreviations: CA19-9, carbohydrate antigen 19-9; N+, lymphnode positivity; CT adj, adjuvant chemotherapy; and G2-3, tumour grading.

**Table 5 cancers-14-01680-t005:** Multivariate analysis for survival prognostic factors.

	HR	L 95% CI	U 95% CI	*p* Value
RM	1.65	0.69	3.92	0.259
DM ± RM	2.58	1.02	6.48	0.044
CA19-9	4.30	1.20	15.38	0.024
N+	1.24	0.57	2.70	0.584
Bismuth IV	5.92	1.87	18.73	0.002
Vascular Invasion	2.51	1.05	6.02	0.039
Complications	4.47	0.86	23.22	0.075
Recurrence	6.91	1.60	29.83	0.009

Abbreviations: CA19-9, carbohydrate antigen 19-9 and N+, lymphnode positivity.

**Table 6 cancers-14-01680-t006:** Kaplan-Meier recurrence analysis.

	R0	R1	RM	DM ± RM
1 year	17.9%	37.2%	35.3%	39.1%
3 years	53.1%	73.6%	64.7%	84.8%
5 years	65.6%	86.4%	73.5%	100%
Median DFS, months	30	16.3	20	15

Abbreviations: DFS, disease-free survival.

**Table 7 cancers-14-01680-t007:** Univariate analysis for recurrence prognostic factors.

	HR	L 95% CI	U 95% CI	*p*-Value
RM	1.32	0.66	2.61	0.432
DM ± RM	2.72	1.39	5.34	0.004
CA19-9 >40 kU/L	2.44	1.13	5.28	0.023
N+	1.52	0.84	2.76	0.169
Bismuth IV	1.50	0.73	3.10	0.274
Dimension	1.00	0.99	1.02	0.547
CT adj	0.97	0.54	1.76	0.925
Stage III-IV	1.51	0.86	2.68	0.154
G2	1.28	0.52	3.11	0.592
G3	1.67	0.61	4.55	0.317
Vascular Invasion	1.97	1.00	3.86	0.049
Complications	3.00	1.18	7.65	0.021

Abbreviations: CA19-9, carbohydrate antigen 19-9; N+, lymphnode positivity; CT adj, adjuvant chemotherapy; and G2-3, tumour grading.

**Table 8 cancers-14-01680-t008:** Multivariate analysis for recurrence prognostic factors.

	HR	L 95% CI	U 95% CI	*p*-Value
RM	1.48	0.66	3.30	0.340
DM ± RM	2.95	1.30	6.69	0.009
CA19-9	1.80	0.62	5.26	0.280
N+	1.01	0.50	2.05	0.973
Bismuth IV	2.07	0.81	5.32	0.130
Vascular Invasion	1.75	0.80	3.85	0.162
Complications	4.20	1.08	16.43	0.039

Abbreviations: CA19-9, carbohydrate antigen 19-9 and N+, lymphnode positivity.

## Data Availability

The data presented in this study are available on request from the corresponding author. The data are not publicly available due to privacy restrictions according to Italian law.

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
