# Peer review of "Impact of Positive Radial Margin on Recurrence and Survival in Perihilar Cholangiocarcinoma"

_cancers, 2022, doi:10.3390/cancers14071680_

Round 1

Reviewer 1 Report

Dear Editor, thank you so much for inviting me to revise this manuscript. This study addresses a current topic.

The manuscript is quite well written and organized. 

Figures and tables are comprehensive and clear.

The introduction explains in a clear and coherent manner the background of this study.

We suggest the following modifications:

  • Introduction section: although the authors correctly included important papers in this setting, we believe some studies regarding adjuvant therapy should be cited within the introduction (PMID: 33571059; PMID: 33307876; PMID: 31986437), only for a matter of consistency. We think it might be useful to introduce the topic of this interesting study.
  • Methods and Statistical Analysis: nothing to add.
  • Discussion section: Very interesting and timely discussion. Of note, the authors should expand the Discussion section, including a more personal perspective to reflect on. For example, they could answer the following questions – in order to facilitate the understanding of this complex topic to readers: what potential does this study hold? What are the knowledge gaps and how do researchers tackle them? How do you see this area unfolding in the next 5 years? We think it would be extremely interesting for the readers.

However, we think the authors should be acknowledged for their work. In fact, they correctly addressed an important topic, the methods sound good and their discussion is well balanced.

One additional little flaw: the authors could better explain the limitations of their work, in the last part of the Discussion.

We believe this article is suitable for publication in the journal although some revisions are needed. The main strengths of this paper are that it addresses an interesting and very timely question and provides a clear answer, with some limitations.

We suggest the addition of some references for a matter of consistency. Moreover, the authors should better clarify some points.

Author Response

Dear Editor,

we thank the reviewer for the analysis of our paper and for the kind and constructive comments. We are happy to find his/her support for publication of this difficult work.

Schematically we improved the sections as follow:

  • Introduction: we updated the references to have a better background
  • Discussion: we tried to improve it expecially regarding adjuvant therapies and chemotherapy administration as suggested by another reviewer. During this explanation we hope we could improve our belief in the natural history of RM positivity, and the limits of our work.

We hope that this revised manuscript will give the reviewer a better understandings of our belief and point of view of this disease

Reviewer 2 Report

I have no concerns about this manuscript, that is well written as well as really interesting to readers.

Best regards

Author Response

We thank the reviewer for supporting our work

Reviewer 3 Report

This paper, while asking interesting questions about which margins are of clinical significance in the setting of resected hilar cholangiocarcinoma, is quite flawed.

To begin with, the authors do not cite nor make reference to the critical prospective phase 2 trial, SWOG 0809 (Ben-Josef et al., JCO 2014).  This trial has across many geographies set the standard of care for resected hilar cholangiocarcinoma (who constituted a plurality of patients in this trial), with adjuvant chemotherapy followed by adjuvant chemoradiation.  The authors of the current series do not cite this trial not make any reference to the role of radiation therapy, which is often used in this setting and others to improve disease-related outcomes when adverse pathologic features are noted, particular R1 resection (as in the SWOG 0809 trial).  Consequently, the interpretability of the authors' data absent this key component of the treatment algorithm is limited.  

In addition, the authors have highly heterogeneous cohorts within this single institution series, with 63% of R0 patients getting adjuvant chemotherapy, compared with 88% for RM patients and 47% for DM+/-RM patients (p=0.039).  This could easily explain any of the observed outcome differences in this series, and the absence of a 'signal' for adjuvant chemotherapy on univariate analysis should not undercut the possible benefit of adjuvant systemic therapy in this setting given the small sample size, likely benefit (HR 0.71) but with limited statistical power, and supported benefit of adjuvant systemic therapy as per the BILCAP trial, for instance (curiously, the BILCAP trial is not cited either).

Putting the pieces together, the authors are justifiably attempting to understand the differential role of radial versus ductal margin status for resected hilar CCA.  However, the heterogeneity of their cohort, particularly with regard to differential receipt of systemic therapy, and total absence of adjuvant radiotherapy, make interpretation of the results exceedingly challenging.  

Author Response

Dear Editor,

we thank the reviewer for the analysis of our paper and for the pointed recommendations to improve it. We thank him/her in particular for raising the point of adjuvant chemotherapy after curative intent surgery, which is in our mind a very important issue.
In detail we tried to improve the manuscript as follow:

-       the Introduction section was expanded, including information and literature references about adjuvant therapies in particular Radiotherapy. 
-       the information regarding adjuvant radiotherapy administration in our study population were checked and updated. In the Discussion section we made a comment on RT patient-based indication process in our centre and its consequent limits
-   the Discussion regarding adjuvant chemotherapy in resected PHC was revised, hopefully obtaining a clearer exposition. We aimed to underline our belief in the role of adjuvant chemotherapy despite the results of our univariate analysis, most likely biased by the small sample size of the series. Indeed we strongly think that systemic chemotherapy is potentially a game changer in this disease but most patients cannot get it due to postoperative complications, jaundice and infective complication after biliary drainage. In our mind patients with DM recurrence after curative intent surgery are likely to develop  anastomotic complications and recurrent obstructive  jaundice, while patients with isolated RM positivity are more likely to develop systemic recurrence. The local recurrence has definitively less chance to be treated with chemotherapy due to the factor cited above, while patients with systemic recurrence (eg lung or bone metastases) could be more likely treated. We could not prove clearly this speculation but we think that the cohorts are eterogeneous in chemotherapy administration for this reason.

Round 2

Reviewer 1 Report

The authors modified the manuscript according to our suggestions. We recommend Acceptance.